# DEBOSH: Deep Bayesian Shape Optimization

## Abstract

Graph Neural Networks (GNNs) can predict the performance of an industrial design quickly and accurately and be used to optimize its shape effectively. However, to fully explore the shape space, one must often consider shapes deviating significantly from the training set. For these, GNN predictions become unreliable, something that is often ignored. For optimization techniques relying on Gaussian Processes, Bayesian Optimization (BO) addresses this issue by exploiting their ability to assess their own accuracy. Unfortunately, this is harder to do when using neural networks because standard approaches to estimating their uncertainty can entail high computational loads and reduced model accuracy. Hence, we propose a novel uncertainty-based method tailored to shape optimization. It enables effective BO and increases the quality of the resulting shapes beyond that of state-of-the-art approaches.

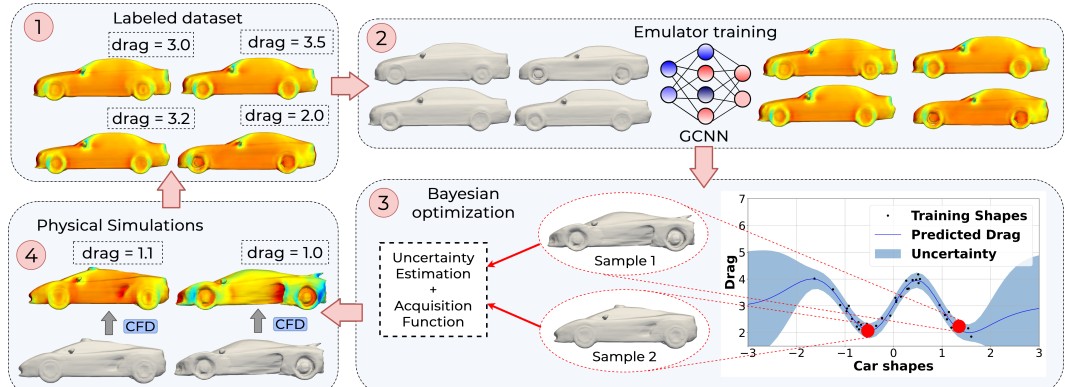

Figure 1: **_DEBOSH_ pipeline. (1)** Run physical simulations. **(2)** Train the GNN. **(3)** Evaluate the acquisition function on samples without an associated physical simulation. **(4)** Select promising samples according the acquisition function, optimize their shape, add them to the training set, and go back to step 1.

# 1 Introduction

Computational Fluid Dynamics (CFD) simulations are key to maximizing the performance of aircraft wings (Li et al., 2019), windmill blades (Jureczko et al., 2005), hydrofoils (Ching-Yeh et al., 2006), car bodies (Liu, 2008), ship hulls (Dejhalla et al., 2001), and propellers (Gur & Rosen, 2009). However, because potentially expensive simulations must be run for each design change, a widespread engineering practice is to test only a few. One way to actually explore the shape space is to use genetic algorithms (Gosselin et al., 2009). In their simplest form, they require many evaluations of a fitness function and, therefore, many expensive simulations, which makes them inefficient. Alternatives include topology optimization (Saviers et al., 2019) and adjoint differentiation (Allaire, 2015; Gao et al., 2017; Behrou et al., 2019). The first is highly effective but only applicable in very specific cases. The second estimates gradients of the fitness function with respect to deformations of the 3D mesh and is only truly applicable to relatively small deformations.

To reduce the computational cost, surrogate models are often used instead. Given a set of parameterized shapes for which simulations are available, non-linear interpolation is used to predict the performance of shapes whose parameters are different, which makes it possible to optimize with respect to the parameters without further simulations. The most popular such method is known as *Kriging* (Laurenceau et al., 2010) and relies on Gaussian-processes (GP) to perform the interpolation (Rasmussen & Williams, 2006). A strength of GPs is that they provide not only estimates of the quality of any given shape but also the reliability of that estimate. This enables Bayesian Optimization (BO) (Mockus, 2012): Given a large set of shapes and assuming that simulations are available only for a subset of these, BO finds the best compromise between performing a search for optimal shapes in regions where the model is certain about its predictions and exploring areas where the model is uncertain and good shapes could be found, even though their predicted performance is low.

Unfortunately, Kriging works best for models that can be parameterized using relatively few parameters, which limits its applicability. Hence, Graph Neural Networks (GNNs) (Boscaini et al., 2016; Monti et al., 2017) have emerged as an alternative way to formulate surrogate models (Baqué et al., 2018; Hines & Bekemeyer, 2023). Given a collection of 3D surface meshes and corresponding simulation results, they can be trained to emulate a complex fluid-dynamics simulator and can handle models with arbitrarily large numbers of parameters. A key limitation, however, is that, unlike GPs, GNNs do not provide an estimate of the reliability of their predictions, which precludes Bayesian Optimization.

In this paper, we introduce *Deep Bayesian Shape Optimization* (*DEBOSH*) to overcome this limitation: We first use existing approaches–Deep Ensembles (Lakshminarayanan et al., 2017) and MC-Dropout (Gal & Ghahramani, 2016)–to estimate the uncertainty of our GNNs and to incorporate them into a Bayesian Optimization framework, which, surprisingly, had not been done before. As depicted by Fig. 1, our method entails an iterative process that encompasses surrogate model training, the exploration of shapes for which simulations have not yet been conducted, guided by both surrogate model predictions and their associated uncertainty, and the execution of simulations for selected shapes.

When using Ensembles, *DEBOSH* delivers good results but still at a steep computational cost. Thus, we propose to replace Ensembles by *Reentrant GNNs* that deliver an even better accuracy by exploiting a key property of CFD computations: The performance values—drag for cars, lift-to-drag ratio for planes—we want to estimate can, in theory, be computed by integrating pressure values along the surface. Thus, as shown in Fig. 2, we make our GNNs estimate both the performance value and the pressure fields. We then iteratively feed back the pressures for increasingly accurate performance estimates. In essence, when predicting performance, the network has access to rough pressure estimates across the *whole surface*, which makes it possible to account for non-local effects. Not only does this increase accuracy, but it also gives rise to a useful behavior: For in-distribution validation samples, we see rapid convergence of the performance estimates towards an usually correct value. For out-of-distribution ones, the convergence is much slower, and the limit is often wrong. In other words, convergence speed can be used as a proxy for the reliability of the estimates and can be computed by training a *single* network.

In short, our contribution is both an effective BO framework that relies on deep-learning-based surrogate models and a new deep-learning architecture, the reentrant GNN, that is particularly well suited for it. We demonstrate its superior performance when optimizing the shapes of 2D airfoils and 3D cars. Our code and training data will be made publicly available.

## 2  Related Work

In all engineering fields that involve running computationally demanding simulations to estimate the performance of 3D shapes, maximizing this performance is often difficult. First, it usually is a highly non-convex function of the design parameters. Second, in industrial practice, the simulator can only be run so many times, which limits how thoroughly the design space can be explored. In this section, we briefly review some of the dominant approaches to addressing these issues.

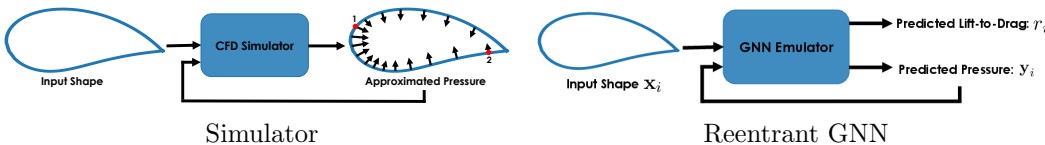

Figure 2: **From simulator to reentrant GNN. (Simulator)** Consider distant airfoil locations 1 and 2. In a traditional simulator, results at location 1 can easily influence the results at location 2 in spite of their distance, as often happens in real life. **(Reentrant GNN)** The recursive nature of the computation makes it possible to take into account all the physical parameter values while estimating one at a specific location, hence making the interaction between distant locations straightforward.

## 2.1 SHAPE OPTIMIZATION

A popular way to automate shape optimization is to use genetic algorithms (Gosselin et al., 2009). However, as they require many evaluations of the fitness function, which involves running an expensive physical simulation, a naïve implementation would be inefficient. *Kriging*, or Gaussian-process (GP) regression (Rasmussen & Williams, 2006), is one of the most popular ways to reduce the required number of evaluations (Jeong et al., 2005; Laurenceau et al., 2010; Toal & Keane, 2011; Xu et al., 2017; Umetani & Bickel, 2018). The measures of uncertainty delivered by Gaussian processes, in addition to their predictions, can be used to explore the shape space more effectively. This works for models controlled by relatively few design parameters but does not scale well to high-dimensional inputs. One must instead rely on low-dimensional shape parameterizations that can be difficult to design and often prevent the full exploration of the shape space.

Convolutional Neural Networks (CNNs) running on 3D voxel grids can be used to remedy this. It has been done to accelerate the search for solutions of the discrete Poisson equation (Tompson et al., 2017), to directly regress the fluid velocity fields given an implicit surface description (Guo et al., 2016), or to compute fluid simulations velocities from a set of reduced parameters (Kim et al., 2019). However, because the underlying 3D CNN architectures have to work on 3D grids, they tend to suffer from large memory footprints and computationally demanding inference. This can be improved by replacing the CNNs by Graph Neural Networks (GNNs) (Boscaini et al., 2016; Monti et al., 2017) operating on 3D meshes (Baqué et al., 2018; Remelli et al., 2020; Hines & Bekemeyer, 2023). Given a set of 3D meshes, the GNN is trained to predict their aerodynamic characteristics, as computed by standard CFD packages (Drela, 1989; Weller et al., 1998; Mountrakis et al., 2015). These characteristics are then used to write an objective function that is differentiable with respect to the shape parameters. The objective function can then be minimized with respect to these parameters.

## 2.2 UNCERTAINTY ESTIMATION

The techniques discussed above allow the refinement of complex shapes parameterized by large vectors of design variables. Because CNNs and GNNs are differentiable, this can be done using a gradient-based technique. However, such techniques can easily be caught in local maxima. In this work, we incorporate them into Bayesian Optimization (BO) (Mockus, 2012). It is one of the best-known approaches to finding global minima of a black-box function $g : \mathbf{A} \rightarrow \mathbb{R}$, where $\mathbf{A}$ represents the space of possible shapes, without assuming any specific functional form for $g$. For our purposes, $g$ is a GNN.

As described in more detail in Appendix A.2, BO relies on an *acquisition function* to gauge how desirable it is to evaluate a point, given the current model state. It is a function of the values predicted by the model and their associated uncertainty and designed to favor samples with the greatest potential for improvement, potentially over the current optimum (Qin et al., 2017; Auer, 2002). Hence, the model must be able to deliver a reliability estimate for its predictions, something that CNNs and GNNs do not naturally do.

MC-Dropout (Gal & Ghahramani, 2016) and Deep Ensembles (Lakshminarayanan et al., 2017) have emerged as two of the most popular approaches to remedying that; with Bayesian

networks (Mackay, 1995) being a third alternative that is often more difficult to train effectively (Ashukha et al., 2020). MC-Dropout involves randomly zeroing out network weights and assessing the effect, whereas Ensembles involve training multiple networks, starting from different initial conditions. In practice, the latter tends to perform better but can be much more computationally demanding, chiefly because the training procedure has to be restarted from scratch multiple times.

An alternative is to use sampling-free methods that estimate uncertainty in one single forward pass of a single neural network, thereby avoiding computational overheads (Amersfoort et al., 2020; Malinin & Gales, 2018; Tagasovska & Lopez-Paz, 2018; Postels et al., 2019). However, deploying them usually requires heavily modifying the network's architecture (Postels et al., 2019), significantly changing the training procedures (Malinin & Gales, 2018), or limiting oneself to very specific tasks (Amersfoort et al., 2020; Malinin & Gales, 2018; Mukhoti et al., 2021). Additionally, using these methods can result in reduced prediction accuracy (Postels et al., 2022). We will show that our Reentrant GNNs do not suffer from these drawbacks.

## 3 Method

Given a small training dataset of shapes with simulated physical performance data and a much larger pool of shapes without any simulations, *DEBOSH* iteratively repeats the following four steps depicted by Fig. 1:

1. Use the training shapes and simulation results to train a surrogate model $\widetilde{g}_\Theta$.
2. Take each shape from the unlabelled pool and make a prediction with $\widetilde{g}_\Theta$.
3. Given the uncertainty of the predictions, compute the acquisition function introduced in Section 2.2 for the shapes in the unlabelled pool.
4. Pick the best new shapes in terms of the acquisition function, optimize their shape with gradient optimization, add them to the training set, and iterate.

These are standard BO steps, as described in Appendix A.2, except for step #4. It involves exploring the shape space without running additional simulations. It takes advantage of the fact that GNNs allow for gradient-based shape optimization. The key to implementing *DEBOSH* is an effective way to estimate not only the performance value associated with a shape but also the uncertainty on this estimate in step #3, which is something ordinary GNNs (Monti et al., 2017) do not provide.

### 3.1 Formalization

Given a set of $N$ 3D shapes $\{\mathbf{x}_i\}_{1 \leq i \leq N}$ represented by triangulated meshes, we run a physics-based simulator yielding a corresponding set $\{\mathbf{y}_i\}_{1 \leq i \leq N}$ of physical values, such as pressure at each vertex. Let $R$ be the function that takes as input the $\mathbf{y}$ values and returns an overall performance value $r = R(\mathbf{y})$, such as overall drag for a car or lift for a wing. $R$ is task-specific. For example, in the case of drag, it is computed by integrating pressure values over the 3D shape. Assuming that each mesh $\mathbf{x}_i$ is parameterized by a lower-dimensional latent vector $\mathbf{z}_i$ and that there is a differentiable mapping $\mathbf{P} : \mathbf{z} \rightarrow \mathbf{x}$, this gives us the initial training set $T = \{(\mathbf{z}_i, \mathbf{x}_i, r_i, \mathbf{y}_i)\}_i$ that we need to initialize our optimization scheme. Similarly, we expect a larger pool of unlabeled shapes, consisting of latent vectors and meshes denoted as $U = \{(\mathbf{z}_i, \mathbf{x}_i)\}_i$, but no simulation data. We train the surrogate model using samples from $T$ (Step 1) and use it to perform predictions (Step 2), compute the acquisition function (Step 3), and select samples for simulations from the set $U$ (Step 4).

### 3.2 Using a Standard GNN

*DEBOSH* relies on a GNN-based surrogate model to emulate physical simulations and return performance values for 3D shapes. To this end, we use the GNN of (Baqué et al., 2018) to predict the $\mathbf{y}$ values for a given shape $\mathbf{x}$, from which we can infer the overall performance

value $r = R(\mathbf{y})$. In practice, the GNN can also be trained to predict both $r$ and $\mathbf{y}$ from $\mathbf{x}$, which turns out to be more effective. We revisit this issue in Section 3.3.

As mentioned in Section 2.2, one way to estimate the reliability of these predictions is to use Ensembles. That is, at each training iteration—step 1 of the *DEBOSH* algorithm—we can train several GNNs and use the variance of their predictions as an uncertainty estimate to evaluate the acquisition function. Unfortunately, this is computationally demanding. Even though MC-Dropout (Gal & Ghahramani, 2016) is a less demanding alternative, for our purposes, it delivers worse uncertainty estimates and lower overall performance, which is consistent with what has been reported elsewhere (Ashukha et al., 2020).

### 3.3 Using a Reentrant GNNs

Our experiments show that using Ensembles, as discussed above, outperforms standard approaches, which is one of the contributions of the paper. However, this comes at a cost because training ensembles is expensive. We now show that we can do better in terms of both accuracy and computational complexity by designing a special-purpose architecture that accounts for the specificities of the simulators our GNNs are designed to emulate.

**Motivation.** Traditional simulators rely on solving differential equations, such as the Navier-Stokes equations. To this end, they compute a sequence of approximate solutions, and each one is used to refine the next. This plays a role in modeling non-local interactions in which a local part of the shape $\mathbf{x}$ can influence the simulation output $\mathbf{y}$ at a distant location, as depicted by Fig. 2(Simulator). By contrast, in the GNN of (Baqué et al., 2018), information propagation across the shape happens at the pace of successive convolutions. Hence, it is comparatively slow. Hence, even when there are many convolutional layers, information may not be transmitted across the whole surface during a single forward pass.

Furthermore, $r$ and $\mathbf{y}$ are connected through integration function $r = R(\mathbf{y})$. For example, when optimizing an airfoil, $\mathbf{y}$ represents pressure for every vertex, and $r$ is the lift-to-drag value. As shown in Appendix A.4, computing $\tilde{r}$ from the predicted $\tilde{\mathbf{y}}$ analytically is less accurate than training the network to predict it directly. A way to resolve this issue is to learn a mapping $\tilde{\mathbf{y}} \to \tilde{r}$, which is what our proposed architecture does.

**Reentrant Architecture.** Thus, we introduce the *Reentrant GNN* depicted on the right of Fig. 2. They iterate $I$ times. At iteration $i$, they take as input the shape $\mathbf{x}$ and the current estimate of physical properties $\tilde{\mathbf{y}}^{i-1}$ and return an updated vector $\tilde{\mathbf{y}}^i$, along with a new performance estimate $\tilde{r}^i$. Initially, we take $\tilde{\mathbf{y}}^0$ to be uniformly zero. For each training batch, we randomly pick the number of iterations $I$ between 1 and a fixed number $M$. We supervise the final predictions $\tilde{\mathbf{y}}^I$ and $\tilde{r}^I$ with ground truth targets $\mathbf{y}$ and $r$.

This improves on standard GNNs in two important ways: First, the successive iterations approximate better the behavior of a CFD simulator. Second, reentrant GNN estimates individual components of $\tilde{\mathbf{y}}$, given estimates obtained at the previous iteration for all components, which means it can account for the non-local effects mentioned above. Our experiments confirm that our recursive approach yields more accurate predictions.

This is in the same spirit as the stacked hourglass networks for pose estimation (Newell et al., 2016) or the nested U-Net (Zhou et al., 2018) for image segmentation. However, unlike these, our network reuses its own outputs as inputs. This allows them to expand their receptive field faster than traditional GNNs, as discussed in Appendix A.4. They use the $\tilde{\mathbf{y}}^i$ estimates for increasingly accurate estimations of $\tilde{r}^i$. This exploits the specificities of our problem because, as discussed in Section 3.2, in theory, the $\tilde{r}^i$ should be computable from the $\tilde{\mathbf{y}}^i$ by integration. In a sense, our network learns to integrate, as shown in Appendix A.5.

**Uncertainty Estimation.** In addition to improving accuracy, our reentrant GNNs provide us with an effective way to estimate uncertainty: In the experiments reported in the following section, given a specific distribution of training samples, for in-distribution validation samples, we observe rapid convergence of the successive $\tilde{r}^i$ towards a $\tilde{r}^{lim}$ value, which is usually correct. By contrast, for out-of-distribution samples, we still observe convergence but towards a value $\tilde{r}^{lim}$ that is often wrong, but at a much slower rate and with oscillations, as shown in Fig. 3. Interestingly, when using true physics-based iterative solvers, a similar

phenomenon can be observed. Their convergence rate often depends on the complexity of the shape given as input (McAdams et al., 2011; Fedkiw et al., 2001).

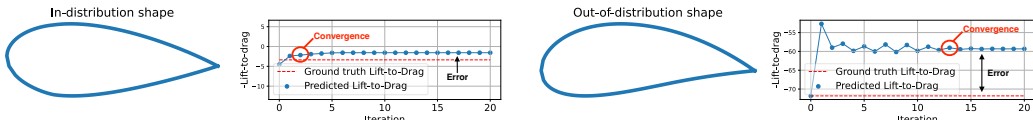

Figure 3: **Convergence rate vs error.** For the in-distribution airfoil to the left, the consecutive values of $\tilde{r}^i$ converge quickly and the limit is very close to the correct answer. By contrast, for an out-of-distribution airfoil, the convergence is much slower and the limit is wrong. This is a behavior that we have consistently observed in our experiments.

To understand why this is the case, in Appendix A.1, we analyze the behavior of a simple Reentrant Multi-Layer Perception (MLP) applied to a 1D case: For any sample $x$ within the training sample distribution, the network is trained to produce the correct $\tilde{r}^i$ for any value of $i$ between 1 and $M$, the maximum number of iterations. By contrast, for a sample $x$ out-of-domain, because deep networks are often bad at extrapolating, there is no particular reason for successive values of $\tilde{r}^i$ to be similar and they are not. The role of the second argument of $f$ can also be understood as that of a prompt sent to the network. For values of $x$ and $\tilde{\mathbf{y}}^i$ within the training distribution, the prompt helps produce the right answer. Taking the prompt to be the output of the previous iteration helps ensure that it is indeed in distribution. For $x$ out-of-domain, there is no reason for this to be so. What we observe is convergence towards a random fixed point.

Thus, we use the rate of convergence of $\tilde{r}^i$ values as a proxy for accuracy. More specifically, we count how many iterations it takes for the difference between $\tilde{r}^i$ and $\tilde{r}^{i-1}$ to drop below a threshold $\delta$. More formally, we take the uncertainty $\sigma(\mathbf{x})$, final prediction $\mu(\mathbf{x})$, and the acquisition function $a(\mathbf{x})$ used in Step 3 of the *DEBOSH* algorithm to be

$$
\begin{aligned}
\mu(\mathbf{x}) &= \tilde{r}^i \ , \\
\sigma(\mathbf{x}) &= i \ , \\
a(\mathbf{x}) &= \mu(\mathbf{x}) + \lambda \sigma(\mathbf{x}) \ ,
\end{aligned}
\tag{1}
$$

where $i$ is such that $\forall j < i, \|\tilde{r}^j - \tilde{r}^{j-1}\| > \delta$ and $\|\tilde{r}^i - \tilde{r}^{i-1}\| < \delta$, and $\lambda > 0$ is a hyperparameter that controls exploitation-vs-exploration tradeoff. Our experiments demonstrate empirically that $\sigma(\mathbf{x})$ is a valid uncertainty measure for the predicted $\mu(\mathbf{x})$ in the following sense: If we train a GNN to predict performance values using samples that come from a particular distribution, its predictions will have lower uncertainty for previously unseen samples from the same distribution than for out-of-distribution samples.

## 4 EXPERIMENTS

We now compare *DEBOSH* against state-of-the-art alternatives in two application scenarios, optimizing airfoils in 2D and car shapes in 3D.

### 4.1 BASELINES

We compare against the following baselines:

**KNN**: Given a set of simulated shapes, we use a standard K-Nearest Neighbors regressor to estimate the performance of additional shapes and add the best one to the training set.

**Kriging**: Using a Gaussian Processes (GPs) to estimate performance values and corresponding uncertainty (Laurenceau et al., 2010). As discussed in Section 2.1, it can be directly used to perform Bayesian Optimization.

**GNN**: GNNs (Baqué et al., 2018; Hines & Bekemeyer, 2023) are a valid alternative to GPs for the purpose of estimating performance numbers. Since they do not compute

uncertainties, we simply add the ones that receive the best score from the GNN to the training set and optimize their shape as in (Baqué et al., 2018).

**DEBOSH/Ens**: We use sets of GNNs to predict mean and variances of performance values, which is known as an Ensemble-based technique. These are then exploited by the *DEBOSH* procedure introduced at the beginning of Section 3.

**DEBOSH/Drp**: Instead of using Ensembles to estimate the performance numbers and their uncertainty, we use MC-Dropout in the *DEBOSH* procedure.

**DEBOSH/Full**: Using the *Reentrant GNN* of Section 3.3 to estimate the performance numbers and their uncertainty in the *DEBOSH* procedure.

We will make the code publicly available to allow reproduction of our experiments.

### 4.2 AIRFOIL OPTIMIZATION IN 2D

2D airfoil profile optimization has become a *de facto* standard for benchmarking shape optimization in the CFD community. In industrial practice, profiles have long been parameterized using a three-dimensional NACA parameter vector (Baqué et al., 2018) and optimized using conventional Kriging-based methods (Jeong et al., 2005; Chiplunkar et al., 2017). For this experiment, we generated an initial dataset of 1500 shapes for airfoil optimization by randomly selecting NACA parameters $\mathbf{z}_i$ and then producing corresponding 2D contours $\mathbf{x}_i$ for each one.

We use the popular XFoil simulator (Drela, 1989) to compute the pressure distribution $\mathbf{y}_i$ over the surface of each $\mathbf{x}_i$. Even though GNN models are primarily designed to handle 3D shapes, they can also handle 2D ones by considering the 2D equivalent of a surface mesh, which is a discretized 2D contour. As in (Baqué et al., 2018), we train our model to predict pressure values $\tilde{\mathbf{y}}_i$ at each vertex along with the global lift-to-drag ratio $\tilde{\mathbf{r}}_i = R(\mathbf{y}_i)$. This is a standard measure of aerodynamic efficiency in a given flight configuration. It is computed as the ratio of the lift generated by the airfoil moving through the air divided by the aerodynamic drag caused by that motion. We split the data into three groups: 1000 samples for training, 300 for testing, and 200 top-performing shapes that we will treat as out-of-distribution samples when gauging the quality of our uncertainty estimates.

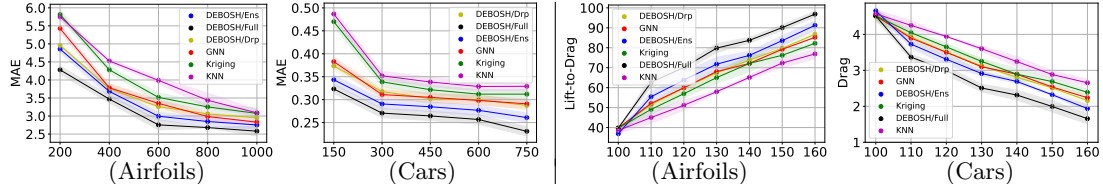

Figure 4: **Left.** Accuracy of the lift-to-drag estimate as a function of the number of exemplars used to train the emulators. **Right.** Lift-to-drag ratio of the shapes during optimization, as a function of number of iterations.

We use the resulting dataset to compare the six approaches introduced in Section 4.1 in terms of emulator accuracy, shape optimization performance, and uncertainty estimation quality. For *DEBOSH/Ens*, we train five GNNs at each iteration while we train only one for *DEBOSH/Drp*. In both cases, we take the uncertainty to be the variance over 5 predictions. For GPs, we use the squared exponential kernel, which has been shown to be particularly effective for aerodynamic prediction (Toal & Keane, 2011; Rosenbaum & Schulz, 2013). For *KNN*, we utilize $K = 8$ and distance-based neighbor weighting as in (Baqué et al., 2018).

As all six methods being compared rely on an emulator, the left *airfoil* plot in Fig. 4 depicts the accuracy of each on the test set as a function of the number of samples from the training set used to train it. Our reentrant GNN outperforms the others consistently, especially when there are only a few training examples.

To similarly evaluate the quality of our uncertainty estimates, we use the same insight as in (Durasov et al., 2022): A network trained on a set of shapes drawn from a given distribution should be more confident on shapes drawn from the same distribution than

| | Kriging | DEBOSH/Ens | DEBOSH/Drp | DEBOSH | |
|---|---|---|---|---|---|
| ROC-AUC | $0.79 \pm 0.01$ | $\mathbf{0.88 \pm 0.01}$ | $0.84 \pm 0.02$ | $0.87 \pm 0.01$ | AIR |
| PR-AUC | $0.78 \pm 0.01$ | $0.86 \pm 0.02$ | $0.82 \pm 0.01$ | $\mathbf{0.88 \pm 0.01}$ | |
| ROC-AUC | $0.62 \pm 0.02$ | $\mathbf{0.90 \pm 0.02}$ | $0.73 \pm 0.01$ | $0.86 \pm 0.01$ | CAR |
| PR-AUC | $0.52 \pm 0.01$ | $0.78 \pm 0.01$ | $0.62 \pm 0.02$ | $\mathbf{0.79 \pm 0.02}$ | |

Table 1: **Evaluation the uncertainty measure for 2D airfoils (AIR) and 3D cars (CAR).** The best result in each category is in **bold** and the second best is in bold. They all correspond to *DEBOSH* and *DEBOSH/Ens*. The two approaches are comparable in terms of evaluating uncertainty but the reentrant GNNs deliver better accuracy, as shown in Fig. 4.

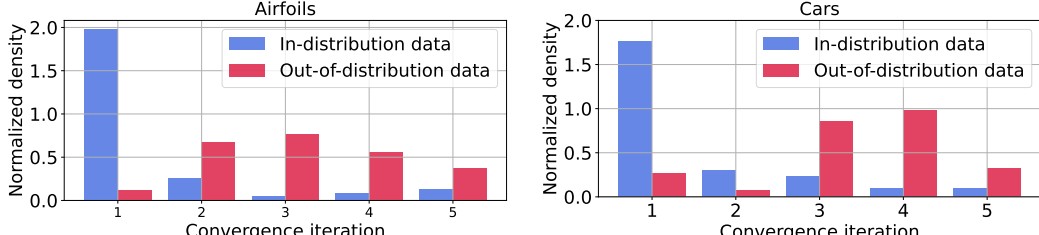

Figure 5: **Convergence rates for in- and out-of-distribution samples.** We plot the distribution of the number of iterations to convergence of our Reentrant GNNs for in-distribution vs. out-of-distribution samples from the test sets of airfoils and cars. In general, convergence takes significantly fewer steps for in-distribution samples than for out-of-distribution ones.

on shapes drawn from a different one. To test this, given all the 3D shapes we have, we took the 200 top-performing ones in terms of their lift-to-drag ratio to be the out-of-distribution samples. The remaining shapes were then considered as the in-distribution ones. One thousand of these were used to train the emulators, and the others were used for testing purposes. After training, we generated uncertainty values for each shape in the in-distribution and out-of-distribution test sets. Finally, we computed standard ROC-AUC, PR-AUC (Malinin & Gales, 2018) metrics for in- or out-of-distribution classification based on the uncertainty estimate. As can be seen in the top rows of Tab. 1, our approach generates uncertainty of a quality similar to that of ensembles, which supports our claim of Section 3.3 that $\sigma(\mathbf{x})$ is valid uncertainty measure. Furthermore, as shown in Fig. 5, our reentrant GNNs often only require 3 iterations for converge. This makes them a little faster than an ensemble of 5 ordinary GNNs and, importantly, requires far less memory and training time. We provide more details in Appendix A.3.

We now turn to shape optimization using each one of the 6 methods. In each case, we used 100 randomly chosen samples from the training set, along with the corresponding simulations, to train the initial emulator. The rest of the training set, plus the OOD set, were treated as a set of unlabelled shapes. After the initial training, we ran the inference for each shape in it. For non-uncertainty approaches (*KNN* and *GNN*), this yielded predicted performance values, and for the other values of the acquisition function (UCB with $\lambda = 3$). We sorted the unlabelled shapes according to these values and picked the 10 best. For GNN-based methods, for each one of these 10 shapes, we also performed 10 steps of gradient-based optimization (Kingma & Ba, 2015). This relatively small number of iterations was chosen to allow us to reap the benefits of GNN-based shape optimization (Baqué et al., 2018), without moving too far away from the starting points and producing shapes whose acquisition value is too different from that of the starting point. We discuss the influence of the number of iterations we perform in Appendix A.6. Finally, we ran simulations for these chosen shapes, added them to the training set, and iterated. For each method, we ran this whole process three times and plot the resulting lift-to-drag ratios as a function of the number of BO iterations performed in the right *airfoil* plot in Fig. 4. The shaded areas depict the corresponding variances. Again, *DEBOSH/Full* outperforms the other approaches by a statistically significant margin.

Recall from Section 3.3 that our approach is predicated on the fact that convergence of the Reentrant GNNs can be expected to be slower for out-of-distribution samples than for

in-distribution ones. The plot on the left side of Fig. 5 validates this hypothesis on the in-distribution and out-of-distribution splits.

### 4.3 Minimizing Car Drag in 3D

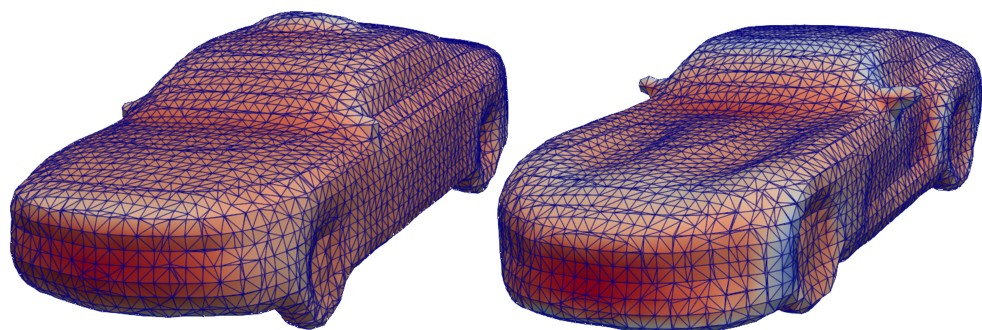

Figure 6: **Optimized car shapes. Left.** A car from the initial training set. **Right.** Best final car. The color depict the vector $\mathbf{y}$ of pressure values predicted by the emulator.

Even though airfoil optimization using the NACA parameterization is a standard benchmark, it relies on a 3D latent vector and is therefore low-dimensional. We now turn to a higher-dimensional problem in which the latent vector is of dimension 256: Car-shape optimization in 3D to minimize aerodynamic drag, as depicted by Fig 6.

We use a cleaned-up and processed subset of the ShapeNet dataset (Chang et al., 2015) that features $N = 1400$ car meshes suitable for CFD simulation. For each such mesh $\mathbf{x}_i$, we run OpenFOAM (Jasak et al., 2007) to estimate the pressure field $\mathbf{y}_i$ created by air traveling at 15 meters per second towards the car. We also use MeshSDF (Remelli et al., 2020) in conjunction with an auto-decoding approach (Park et al., 2019) to learn a function $P : \mathbb{R}^{256} \to \mathbb{R}$ and a set of latent vectors $\{\mathbf{z}_i\}$ such that $\forall i \ \mathbf{x}_i = P(\mathbf{z}_i)$. We chose to use MeshSDF because it yields the differentiable 3D mesh representation we need to optimize with respect to the latent vector components.

As before, we compare all the methods in terms of accuracy, uncertainty, and final performance delivered by Bayesian Optimization. We use the same protocols with one minor modification. For accuracy evaluation, at each iteration, we add 150 new samples. We report our results in the *car* plots of Fig. 4 and the bottom rows of Tab. 1. Again, *DEBOSH/Full* consistently outperforms the other methods. As in the case of airfoils, the plot on the right side of Fig. 5 validates shows that convergence happens faster for in-distribution samples than for out-of-distribution ones.

## 5 Conclusion

We have presented a Bayesian Optimization approach to refining 2D and 3D shapes for increased performance, such as maximizing the lift-to-drag ration of an airfoil or minimizing the aerodynamic drag of a car. It relies on a novel GNN architecture to estimate both the performance values to be improved and the reliability of these estimates. In essence, our GNN learns to both predict physical values, such as pressure, and to integrate them over the whole shape to compute performance numbers. Hence, the estimates it delivers are more accurate than those of competing techniques, which in the end makes it possible optimize shapes better.

In this work, we have focused on aerodynamics but the principle applies to many other devices, ranging from the cooling plates of an electric vehicle battery to the optics of an image acquisition device. In future work, we will therefore explore a broader set of potential applications.

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

# A   APPENDIX

In this section, we first examine the behavior of re-entrant networks in a very simple case. We then provide details about the training procedure and additional supporting evidence for some of the claims made in the paper.

## A.1   ANALYSIS OF A SIMPLE CASE

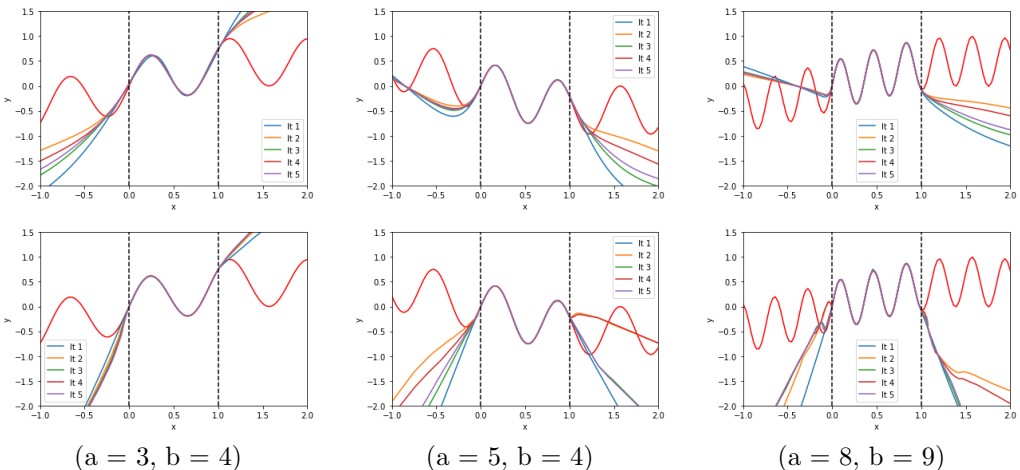

(a = 3, b = 4)            (a = 5, b = 4)            (a = 8, b = 9)

Figure 7: **Learning to interpolate a 1D function**. Using a two-layer perceptron to interpolate $f(x) = \sin(a * x)\cos(b * y)$ given training pairs $(x, f(x))$ for which $0 < x < 1$. Each curve represents the value of $y_i^t$ from Eq. 2 for values of $x$ ranging from -1.0 to 2.0, that is, both inside and outside the training domain. There is one curve per iteration $i$ in Eq. 2, ranging from 1 to 5. **Top row.** Taking tanh to be the activation function. **Bottom row.** Using ReLu.

To model the behavior of our reentrant-GNNs in a simpler and easier-to-analyze context, we replace GNN with a perceptron $f_W$ that takes two scalar inputs $x$ and $y$ and outputs a scalar. Given a training set $\{(x_i, r_i), 1 \le i \le N\}$, we make it re-entrant by computing

$$y_i^1 = f_W(x, 0), \ y_i^2 = f_W(x, y_i^1), \ ..., \ y_i^{(t_i)} = f_W(x, y_i^{(t_{i-1})}) \tag{2}$$

for each $i$, where $t_i$ is a different random integer between 1 and $T$ for each sample. In these examples, we use $T = 5$. We then minimize the total loss $\sum_i (r_i - y_i^{(t_i)})^2$. Fig. 7 depicts the results of this process when the $x_i$ are uniformly sampled between 0 and 1 and the $r_i$ are taken to be $sin(a * x_i) * cos(b * x_i)$ for different values of $a$ and $b$. For values of $x$ between 0 and 1, that is, for values that are within the training domain, we have $y_i^0 \approx y_i^1 .... \approx y_i^T \approx r_i$. In contrast, out of domain, that is, outside the range [0,1], this is not true anymore, and we can see strong oscillations of the successive $y_i^t$ values for $1 \le t \le T$. This makes sense because deep networks are known not to extrapolate well. Thus, even though the network is trained to produce similar predictions for all values of $t$ in-domain, the out-of-domain predictions are essentially random, and there is no reason for them to be equal. In the results section, we showed that, for both airfoils and car shapes, out-of-domain values of $x$ tend to produce oscillations and slow convergence. Interestingly, we observe exactly the same behavior on this very simple example, as evidenced by the fact that the curves of Fig. 7 are *not* superposed for $x < 0$ and $x > 1$.

The exact values obtained for these out-of-domain samples are very hard to predict. As can be seen by comparing the two rows of Fig. 7, they depend critically on the chosen activation function, tanh or ReLu in this case. They also depend heavily on how the networks have been initialized, as can be seen in Fig. 8. In one case, we initialized the weights of our perceptrons using normally distributed weights. In the other, we used the slightly more sophisticated Xavier Initialization (Kumar, 2017).

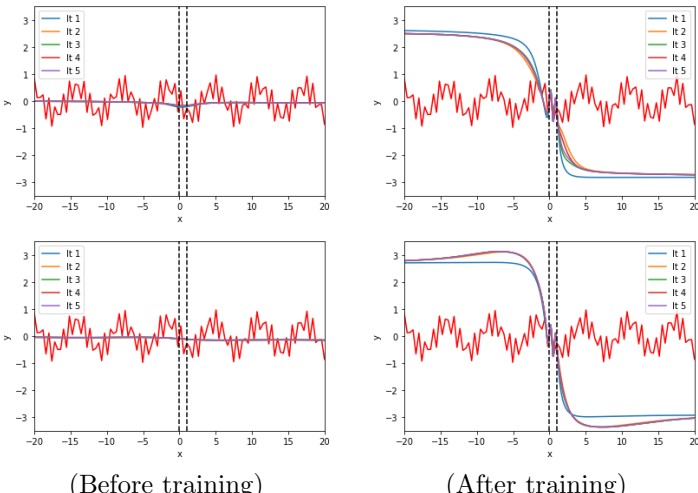

(Before training)                    (After training)

Figure 8: **Influence of initialization.** We plot the same curves as in Fig. 7 when learning to interpolate the function $f(x) = \sin(5*x)\cos(4*y)$, but over a more extended range of $x$ and starting from a different initialization of the perceptron weights in each row. **Before training.** As before, each curve represents the values $y_i^t$ as a function of $x$. Here we plot those returned by our perceptrons after initialization of their activation weights, but before actual training. The two plots correspond to the two different initializations. **After training.** Values after training. There are similar for $0 < x < 1$ but different out of this domain. Not that they are also very different across the two rows because of the slightly different initializations.

Crucially, in all cases, seeing large variations in the values of the successive $y_k(x)$ for a given $x$ is *always* a warning sign that the estimated value is likely to be incorrect. This is what we exploit in this work.

## A.2 BAYESIAN OPTIMIZATION

Given a performance estimator of uknown reliability, exploration-and-exploitation techniques seek to find global optimum of that estimator while at the same time accounting for potential inaccuracies in its predictions.

Bayesian Optimization (BO) (Mockus, 2012) is one of the best-known approaches to finding global minima of a black-box function $g : \mathbf{A} \to \mathbb{R}$, where $\mathbf{A}$ represents the space of possible shapes, without assuming any specific functional form for $g$. It is often preferred to more direct approaches, such as the adjoint method (Allaire, 2015), when $g$ is expensive to evaluate, which often is the case when $g$ is implemented by a physics-based simulator.

BO typically starts with a surrogate model $\widetilde{g}_\Theta : \mathbf{A} \to \mathbb{R}$ whose output depends on a set of parameters $\Theta$. $\widetilde{g}_\Theta$ is assumed to approximate $g$, to be fast to compute, and to be able to evaluate the reliability of its own predictions in terms of a uncertainty. It is used to explore $\mathbf{A}$ quickly in search of a solution of $\mathbf{x}^* = \arg\min_{\mathbf{x} \in \mathbf{A}} g(\mathbf{x})$. Given an initial training set $\{(\mathbf{x}_i, r_i)\}_i$ of input shapes $\mathbf{x}_i$ and outputs $r_i = g(\mathbf{x}_i)$, it iterates the following steps:

> **Step 1:** Find $\Theta$ that yields the best possible prediction by $\widetilde{g}_\Theta$.
>
> **Step 2:** Generate new samples not present in the training set.
>
> **Step 3:** Evaluate an *acquisition function* on these samples.
>
> **Step 4:** Add the best ones to the training set and go back to Step 1.

As shown in the example of Fig. 9, the role of the acquisition function is to gauge how desirable it is to evaluate a point, based on the current state of the model. It is often taken to be the *Expected Improvement* (EI) (Qin et al., 2017) or *Upper Confidence Bound* (UCB) (Auer, 2002) that favor samples with the greatest potential for improvement over

the current optimum. It is computed as a function of the values predicted by the surrogate and their associated uncertainty.

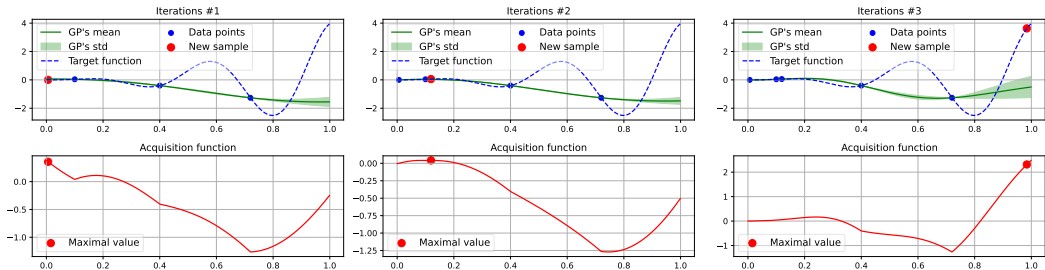

Figure 9: **Bayesian Optimization.** Given three initial data points for the function (dashed blue)we want to optimize, we train a GP surrogate model (Step 1) and compute the UCB acquisition function (Auer, 2002) over the $[0, 1]$ range (Steps 2-3). We then select the points that maximize, evaluate the target function at those points, and include the results in our training dataset (Step 4). The process is then iterated and, eventually, we find the true maximum of the function at $x \approx 1$, whereas a simple gradient based method would probably have remained trapped at the local maximum $x \approx 0.58$.

## A.3 TRAINING SETUPS

| | GNN | DEBOSH/Ens | DEBOSH/Drp | DEBOSH | |
|---|---|---|---|---|---|
| Memory | 1x | 5x | 1x | 1x | AIR |
| Inf. Time | 1x | 5x | 5x | 3x | |
| Train. Time | 1x | 5x | 1x | 2x | |
| Memory | 1x | 5x | 1x | 1x | CAR |
| Inf. Time | 1x | 5x | 5x | 3x | |
| Train. Time | 1x | 5x | 1x | 2x | |

Table 2: **Computational costs.** The Memory, Inference Time, and Training Time metrics measure the amount of time and memory required to train the network(s) and to perform inference, in comparison to a single model.

For our experiments, we used single Tesla V100 GPU with 32Gb of memory. The training process was implemented using the Pytorch (Paszke et al., 2017) and Pytorch Geometrics (Fey & Lenssen, 2019) frameworks.

**Airfoils.** For airfoils, we have generated 1500 shapes from NACA parameters, and simulated pressure and lift-to-drag values with XFOIL simulator. As an emulator, we use architecture that consists of 35 GMM layers (Monti et al., 2017) with ReLU activations. First, we extract node features with these GMM layers and pass them to pressure branch, that consists out of 3 GMM layers, and lift-to-drag branch, that uses global pooling and 3 fully-connected layers to predict final scalar. For training, we use Adam optimizer (Kingma & Ba, 2015) and perform 200 epochs with 128 batch size and 0.001 learning rate. Both for lift-to-drag and pressure, we use mean squared error (MSE) loss and combine them into final loss with weights 1 for scalar and 100 for pressure. $\delta$ value for convergence method is set to 0.1

**Cars.** For cars dataset, we have generated 1500 shapes from MeshSDF vectors, and simulated pressure and drag values with OpenFOAM simulator. As an emulator, we use architecture that consists of 50 GMM layers with ELU activations (Clevert et al., 2015) and skip-connections (He et al., 2016). Similar to airfoils, we extract node features with these GMM layers and pass them to pressure branch, that consists out of 5 GMM layers, and drag branch, that uses global pooling and 5 fully-connected layers to predict final scalar. For training, we use Adam optimizer and perform 6 epochs with 8 batch size and 0.001 learning rate. Both for lift-to-drag and pressure, we use mean squared error (MSE) loss and combine them into final loss with weights 1 for scalar and $1/200$ for pressure. $\delta$ value for convergence method is set to 0.05.

### A.4 Propagating Information

In a standard GNN information is propagated across the shape with each successive convolution. Hence, it is comparatively slow and our reentrant GNNs address this. To support, this claim we ran an experiment to test the influence of the receptive fields of the GNNs, which control the speed at which information percolates across the network. We trained 5 airfoils and car emulator models of increasing depth while keeping total weights number fixed. Starting from the original architecture, we plot the prediction mean error for both lift-to-drag and drag in Fig. 10 in red. As expected, the error decreases as depth increases and more information is propagated across the shape. The exact same behavior can be observed when using a reentrant GNN run iteratively, as shown by the black curves. This supports our claim that each iteration helps propagate the information across the shape just as effectively as when using the deeper network.

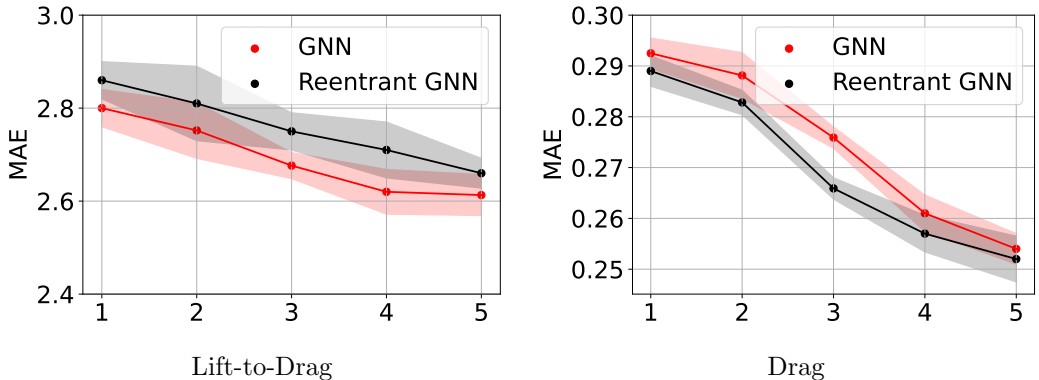

Figure 10: **Propagating information across an shape**. A comparable behavior is observed when increasing the depth of a standard GNN (red curves) and when running several iterations of a shallower reentrant GNN (black curves).

### A.5 Learning to Integrate

We mentioned in Section 3.2 that our reentrant architecture "learns" to integrate the local pressure values to produce a global drag or lift-to-drag ratio. To demonstrate its effectiveness, we compare against several baselines:

1. Only predicting performance value—-drag or lift-to-drag-ration–without predicting the local pressure values.

2. Predicting both performance and local pressure values, but without enforcing consistency, as in (Baqué et al., 2018).

3. Predicting pressure and computing performance using integration.

4. Our recursive approach.

As can be seen in Table 3, our approach does best. Interestingly, training the network to predict the local pressure values even without explicitly using them, as in the #2 approach, helps and yields the second best performing method.

|  | #1: w/o pressure | #2: w/ pressure | #3: integration | #4: *Reentrant* |
|---|---|---|---|---|
| AIR | $3.309 \pm 0.01$ | $2.807 \pm 0.01$ | $3.61 \pm 0.01$ | $2.6 \pm 0.01$ |
| CARS | $0.35 \pm 0.02$ | $0.32 \pm 0.01$ | $0.51 \pm 0.07$ | $0.30 \pm 0.01$ |

Table 3: **Mean error and variance of predicted performance** for the four methods of Section A.5.

## A.6  Gradient Optimization

Given the shapes selected according to the acquisition function during Bayesian Optimization, our method performs several gradient steps in order to refine these shapes and makes them more performant. In this subsection, we examine the impact of performing this optimization.

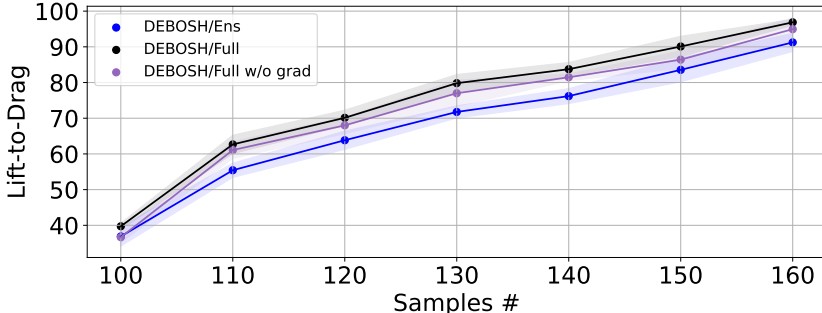

Figure 11: **Impact of refining the shapes.** Turning on gradient optimization of new samples delivers a small performance increase, but smaller than the one used by replacing ensembles with a version of our approach without the refinement.

In the results shown in the main paper, given the current state of the emulator, we performed 10 steps of an Adam-based optimizer with a $1e-4$ learning rate to refine each selected shape. In Fig. 11, we plot the results obtained for the airfoils by doing this refinement (DEBOSH/full), not doing it (DEBOSH/Full w/o grad), or using deep ensembles. DEBOSH without refinement already delivers an improvement overs ensembles, with a further but smaller improvement when performing the refinement. We tried increasing the number of refinement steps but that brought no further improvement.

