# OpenReview forum: "DEBOSH: Deep Bayesian Shape Optimization"
_ICLR.cc/2024/Conference — ICLR 2024 Conference Withdrawn Submission_

### Official Review · Reviewer_UWPo · 2023-10-30

**Soundness:** 2 fair
**Presentation:** 3 good
**Contribution:** 3 good
**Rating:** 5
**Confidence:** 3

**Summary:**

In the traditional industrial design pipeline, the structural design of an object is typically conducted first, followed by physical simulations of the designed structure. Additionally, after each design change, it is necessary to re-run physical simulations. This paper proposes a new paradigm by training a differentiable Graph Neural Network (GNN) using the results of a small number of physical simulations. It further utilizes a Bayesian Optimization approach with uncertainty as the acquisition function to evaluate the results of designs that have not undergone physical simulations and select an optimization direction for design changes. The authors aim to accelerate the entire design pipeline using this approach. The experiments conducted in this study focus on airfoils in 2D and car shapes in 3D.

**Strengths:**

This paper attempts to propose a new pipeline to address the challenges encountered in practical industrial manufacturing, and this research direction is meaningful. Additionally, the authors present a comprehensive technical approach and conduct experiments on relevant data. Overall, the paper is well-written and easy to follow.

**Weaknesses:**

The proposed approach in this paper seems reasonable, but I still have some concerns.

The main issue is that the current design changes are entirely optimized by the network adaptively. While this may indeed yield a higher metric on the designed acquisition function, I am not entirely certain if this metric truly reflects real-world conditions. On one hand, the optimization direction may not necessarily align with the expectations of industrial design (the metric itself may only capture one aspect, while there may be other factors that need to be considered but cannot be formulated). On the other hand, since this metric is also estimated, its accuracy itself deserves further exploration.

Additionally, I have some doubts about step 4 of the method. I am unsure whether incorporating noisy data as supplementary training data would bring benefits or introduce additional noise.

**Questions:**

Firstly, it is necessary to further demonstrate that all metrics in industrial design (at least most of them) can be formulated and that optimization is still feasible when multiple metrics are combined (which would further complicate the optimization landscape).

Secondly, currently, there is only an intuitive explanation regarding the use of uncertainty as the acquisition function, without further analysis, which may be insufficient.

Furthermore, from the perspective of neural networks (without considering analytical solutions), it seems that $\bf{y}$ is equivalent to
$r$. In that case, I would like to know why $r$ is not involved in the optimization process.

Lastly, ablation studies for each aspect need to be supplemented.

I will appropriately increase the score after the aforementioned issues are addressed.

---

### Official Review · Reviewer_ZDCz · 2023-10-30

**Soundness:** 2 fair
**Presentation:** 2 fair
**Contribution:** 2 fair
**Rating:** 3
**Confidence:** 4

**Summary:**

The paper proposes a novel uncertainty-based method tailored to shape optimization.
It enables effective BO and increases the quality of the resulting shapes beyond that of
state-of-the-art approaches.

**Strengths:**

Results are good.
Moderately good description of the process.

Good for CFD community.

**Weaknesses:**

Only one equation in manuscript - indicates lack of soundness and novelty.

Do not agree that use of  Reentrant GNN with Bayesian optimization is sufficient for ICLR level.

Where are failure cases illustrated and discussed ?

**Questions:**

What is "shape optimization" ?

In CS/ML/DL/AI literature - if one searches for that
you get other methods being used/referred.

Specifically say in Vision or Graphics domain, its completely different.
The term should have been defined, and complexity illustrated properly.

Surely, this optimization in CFD is driven by specific targets, performance metrics etc.

---

### Official Review · Reviewer_2niw · 2023-10-31

**Soundness:** 2 fair
**Presentation:** 1 poor
**Contribution:** 1 poor
**Rating:** 3
**Confidence:** 4

**Summary:**

This paper proposes a Bayesian framework for shape optimization that uses a "reentrant" GNN. The reentrant GNN iterates over its own outputs to refine the predictions, and predicts the approximated surface pressure as well as a key metric, such as the drag for cars and lift-to-drag ratio for airfoils. At inference, the slower convergence rate of the reentrant GNN for out-of-distribution shape serves as a proxy to prediction uncertainty.

**Strengths:**

The paper tackles an important topic, namely uncertainty quantification of surrogate models for geometric design optimization. They have conducted a set of experiments on a dataset of airfoil shapes, and car shapes. The car dataset appears to be particularly challenging since the latent geometric parameter is high dimensional. The model with the entrant GNN obtains good experimental results in predicting the drag and lift-to-drag coefficients, especially when there are a few training samples.

**Weaknesses:**

First, the paper is really dense and quite hard to read. I would suggest some rewriting to make it less compact.

Then, I am not convinced by the approach taken by the authors w.r.t the output of the surrogate model. Here, the surrogate model learns to predict both the surface pressure and the metric value. Even though this metric is computed from approximate pressure values with another neural network, it still breaks the physical integration between the pressure field and the different forces. To foster generalization, the GNN should capture the physical phenomenon through the pressure and other volume fields if necessary, and compute the metrics from them through integration. This leads in a trustful and physically-based metric, which is not the case with this method. It is claimed and showed in the appendix that using the integration of the pressure values has a higher MAE for the considered metrics. This is not surprising as regressing a single value is less challenging than predicting the pressure on the whole surface. However, if a surrogate model were to correctly predict the pressure on the surface, then the drag or lift-to-drag would be actually meaningful.

Besides, the reentrant GNN does not provide uncertainties on the predictions, but only a proxy for it. This is a key and crucial difference with the other methods. Techniques such as Deep Ensembles and MC-Dropout provide approximate Bayesian inference in deep gaussian process, while the reentrant GNN does not. If we take twice the same shape, the output will give the same prediction both times.

As a consequence, the motivation behind the reentrant GNN is not transparent to me. It seems to be a rapid and not ingenious solution with a wrong tool.
I tend to agree on the receptive field analysis. A reentrant GNN with $k$-iterations of depth $d$ should have the same receptive field as a simple GNN with depth $d*k$, and a higher receptive field than a simple GNN with the same depth. Still, in Figure 5 it seems that it converges most of the time in 1 step for in-distribution shapes, and thus the gain in performance seems marginal. The same can be deducted from the comparison between reentrant GNN and GNN ensembles which are both very close in prediction performance.

**Questions:**

What are the metrics given in Table 1 ? Could you provide an explanation for them ?

How do you modify the inputs with Deep Ensembles ?

How do the ground truth pressure and predicted pressure compare ? How the pressure evolve over the iterations?

What is the initial performance estimate ?

---

### Official Review · Reviewer_5pXv · 2023-11-02

**Soundness:** 2 fair
**Presentation:** 3 good
**Contribution:** 2 fair
**Rating:** 5
**Confidence:** 4

**Summary:**

This paper proposes a Bayesian shape optimization method in the context of aerodynamics. The surrogate in the BO framework is a Graph Neural Network (GNN) adept at estimating the pressure at the nodes of the mesh as well as predicting the target performance. For the uncertainty estimation, the paper proposes to apply a few existing uncertainty estimation methods (e.g., Deep Ensembles, MC Dropout) as well as a new physically-inspired method that is computationally more efficient and provides better accuracy and better uncertainty.

**Strengths:**

- The interdisciplinary approach to shape optimization at the intersection of engineering design and machine learning is very appealing.

- The idea of physically-driven uncertainty prediction in the context of shape simulation is inspiring.

- The paper is well-written and well-organized.

**Weaknesses:**

- The framework is limited in the sense that it bounds itself to a set of pre-existing shapes in the dataset. In fact, the main question that is addressed is how one can find the most promising shapes to be passed to physical simulator. This is a less strong achievement than a full-fledged BO which could 'synthesis' new shapes.

- The above point begs the question of how the cost of training and optimization are amortized during later inference stages. At least in the case of 1500 airfoils, I feel the brute-force approach of simulating all of them would have been comparable in computational cost to the proposed approach.

- I think the current validation could have been focused on the key idea: physically-based uncertainty estimation. For example, showing that both the physical simulation and the GNN simulate the long-range dependencies similarly (according to Fig. 2).

- The choice of OOD selection based on the final aerodynamic performance seems not to be well grounded. In general, the dataset, at least for airfoils, seems to be pretty uniform (not diverse). This is inferable from the low-dimensional latent space (3D) and also visually from Fig. 3 where the in-distribution and OOD samples don't differ a lot.

- When the computational cost of the proposed method is compared to GNN ensembles, the fact that ensembling is trivially parallelizable should have been discussed.

**Questions:**

- Is there a reason that the size of training set is limited to around 1000-2000? Would the method work if given a dataset an order of magnitude larger?

- Is the evaluation that we see, for example Fig. 4, done via the surrogates or the underlying simulator?

- What it takes to expand the current single-objective BO to a multi-objective one?